# Treatment outcomes in people with diabetes and multidrug-resistant tuberculosis (MDR TB) enrolled in the STREAM clinical trial

Meera Gurumurthy[1]*, Narendran Gopalan[2&], Leena Patel[3&], Andrew Davis[4], Vignes Anand Srinivasalu[2], Shakira Rajaram[5], Ruth Goodall[4], Gay Bronson[3], STREAM Trial Collaboration¶

1 Vital Strategies, Singapore, Singapore, 2 Indian Council of Medical Research – National Institute for Research in Tuberculosis, Chennai, India, 3 Vital Strategies, New York, New York, United States of America, 4 MRC Clinical Trials Unit at UCL, London, United Kingdom, 5 Wits Health Consortium, Johannesburg, South Africa

¶ Membership of STREAM Trial Collaboration is provided in the Acknowledgments section
& These two authors contributed equally
* mgurumurthy@vitalstrategies.org

## Abstract

There is limited evidence on the effect of DM co-morbidity in those undergoing treatment for MDR-TB. We report post-hoc analyses of participants from the STREAM Clinical Trial (Stage 1 and 2 combined). Participants who self-reported diabetes, had random blood glucose ≥200mg/dl at baseline, or reported taking concomitant medication for diabetes were classified as the DM group. In total, 896 (n=84 DM, n=812 non-DM) and 976 (n=87 DM, n=889 non-DM) participants were included respectively in the efficacy and safety analyses reported here. Summary statistics for efficacy and safety outcomes were calculated. Hazard ratios (HR) for time-to-event outcomes were estimated using Cox-proportional hazard models. Compared to the non-DM group, the DM group were significantly older, more likely to be male and had a higher BMI. The DM group experienced a significantly higher proportion of serious adverse events (SAEs) (41% vs. 22%, p<0.001) but was comparable to the non-DM group on all other safety (grade 3-5 adverse events, deaths, unscheduled visits) as well as all efficacy parameters (proportion with unfavourable outcome, proportion FoR, time to FoR and culture conversion) assessed. The STREAM clinical trial experience indicated that it is possible to achieve similar treatment outcomes in people with MDR-TB who have a DM co-morbidity. However, this sub-population experienced more SAEs, underscoring the importance of close monitoring to manage their impact and improve MDR-TB treatment outcomes.

## Introduction

The dual burden of tuberculosis (TB) and diabetes mellitus (DM) is a global public health concern. Globally, over 1.5 million deaths occurred due to TB in 2020 [1] and DM caused an estimated 11.3% of total deaths in 2019 [2]. Both TB and DM are reported among the top 10 causes of death worldwide. [1,3] Further, DM is a known risk-factor for developing

**Data availability statement:** De-identified data from the STREAM trial is now publicly available on the TB-PACTS platform hosted by C-PATH: https://c-path.org/tools-platforms/tb-pacts/. Stage1: StudyID TB-1019 (STREAM-TB) Stage2: StudyID TB-1032 (STREAM2). In order to access these, one must register for with TB-PACTS. The TB-PACTS steering committee is expected to review all user access applications in a timely manner and this may take up to 4 weeks to process.

**Funding:** STREAM stage 1 was funded by the US Agency for International Development (USAID) through the Cooperative Agreement GHN-A-00–08–0004–00. STREAM stage 2 was jointly funded by USAID and Janssen Research & Development. Additional funding for STREAM was provided by the Medical Research Council (MRC) and the UK Department for International Development (DFID) under the MRC/DFID Concordat agreement, which is also part of the EDCTP2 programme supported by the EU. The MRC Clinical Trials Unit at University College London was supported by the MRC (MC_UU_12023/26). There was no additional funding for the secondary analyses reported in this manuscript. The funders had no role in study design, data collection and analysis, decision to publish, or preparation of the manuscript.

**Competing interests:** The authors have declared that no competing interests exist.

TB [4,5] with higher risks in those with poor glycaemic control. [5] DM results in poorer outcomes such as increased TB treatment failure, delayed culture conversion, higher TB recurrence [6,7] and has also been reported as an independent risk factor of multi-drug resistant TB. [7–9]

Multi-drug resistant TB (MDR-TB), defined as TB resistant to the first-line drugs isoniazid and rifampicin, is a serious threat to public health and has complicated the efforts of TB control programs worldwide. Historically, treatment for MDR-TB have been long-drawn out regimens comprising of less potent and tolerable drugs, resulting in lower rates of treatment success. Several of the top ten countries predicted to have the highest number of cases of DM by 2030 are also among those estimated to have highest MDR-TB prevalence.[1,2] DM therefore poses a significant challenge and evidence-based management of MDR-TB and DM is a public health imperative. The effect of DM on MDR-TB has been reported only in a small number of characterised cohorts [10–12] and treatment for MDR-TB is not optimized for various sub-groups or co-morbidities, including DM.[13] Understanding DM as a co-morbidity in MDR-TB has therefore been identified as a priority area of research. [14,15]

Further, there have been reports of poorer treatment outcomes in those with uncontrolled diabetes [16,17] and suggestions that achieving better glucose control may improve outcomes. [18,19] While several studies report the effect of glycaemic control on treatment outcomes in drug-susceptible TB [17,20–26], there have been none to our knowledge in MDR-TB.

Post-hoc analyses of data from STREAM, a phase 3 randomised control trial that evaluated shorter regimens for MDR-TB [27,28] presented an opportunity to evaluate treatment outcomes in a well-characterized cohort of people with MDR-TB and DM. Further, longitudinal glucose measures available from Stage 2 of STREAM allowed us to perform some preliminary evaluation on factors associated with glucose control in people with MDR-TB. We believe that the results reported, and insights offered here can inform better clinical management of MDR-TB and DM comorbidity.

## Methods

### Analysis population

Modified intention to treat (mITT) and safety populations of STREAM Stage 1 (ISRCTN78372190) and Stage 2 (ISRCTN18148631) were included in the post-hoc analyses reported here. mITT population included all participants who were randomized and had a culture that was positive for *M. tuberculosis* at baseline, excluding those in whom baseline isolates were subsequently found to be susceptible to rifampicin or resistant to both fluoroquinolones and aminoglycosides on phenotypic drug-susceptibility testing. Safety population included all who received at least one dose of trial medication.

STREAM Stage 1 randomized 424 adult participants with smear positive pulmonary TB, with evidence of resistance to rifampicin and sensitivity to fluoroquinolones and aminoglycosides, to "Long" and "Short" regimens in a 1:2 ratio from July 2012 through June 2015.[27] The long regimen (approximately 20 months) consisted of medications used in and provided by the National Tuberculosis Programs of the respective countries, based on the 2011 World Health Organization (WHO) guidelines and the short regimen consisted of moxifloxacin, clofazimine, ethambutol, and pyrazinamide administered over a 40-week period, supplemented by kanamycin, isoniazid, and prothionamide in the first 16 weeks[1]. Participants had scheduled weekly visits during the first 4 weeks and thereafter were clinically evaluated at 4-week intervals through week 132; sputum samples for smear and culture were obtained at baseline and at every visit starting from the week 4 visit.

STREAM Stage 2 randomized 588 participants in a 1:2:2:2 ratio to the Stage 1 "Long" regimen (terminated early due to near universal adoption of the shorter regimen by TB programs), 9-month control regimen (Stage 1 "Short" regimen), 9-month "Oral" regimen with bedaquiline, or "Six-month" regimen with bedaquiline and 8 weeks of second-line injectable (also terminated early because of universal adoption of all-oral regimens and recommendations against use of injectables) between March 28, 2016 and Jan 28, 2020. [28] In terms of follow-up, participants were assessed weekly for the first month, then 4-weekly until week 52, 8-weekly until week 84, and 12-weekly thereafter until end of trial follow-up at week 132. Sputum samples for smear and culture were obtained at baseline and at every visit starting from the week 4 visit at least until week 96. Complete blood count and serum chemistry (including blood glucose) were performed at baseline and at every scheduled visit starting at week 4 visit at least until the week 76 visit (primary endpoint). Further details of trial regimens and trial design, including modifications during implementation are described earlier. [28,29]

## Study exposure

Participants who self-reported DM status at enrolment, had random blood glucose ≥200mg/dl at baseline, or reported taking concomitant medication for diabetes were classified as the DM group. Participants whose DM status could not be determined were excluded from the analyses.

## Study outcomes

We report efficacy outcomes as follows: proportions of participants with "unfavourable" and "favourable" status, proportions with Probable or Definite Failure or Recurrence at the time of primary outcome (FoR), and hazard ratios for times to FoR and culture conversion. And "unfavourable" outcome was defined as death, bacteriological failure or recurrence, and major TB treatment change for any reason. "Favourable" outcome was achieved for any participant with a negative culture for Mycobacterium tuberculosis (without a preceding unfavourable outcome) at week 132 for Stage 1, and at week 76 for Stage 2.[27,28]

A "Definite FoR" event required clear bacteriological evidence of failure or relapse (excluding a proven reinfection with exogenous strain of Mycobacterium tuberculosis), and a "Probable FoR" event required some evidence for failure or relapse in the absence of clear bacteriology.[28,30] We also report safety outcomes as follows: proportions of participants with grade 3-5 adverse events (AEs), serious adverse events (SAEs), treatment-related SAEs, deaths as well as proportions of participants with unscheduled visits, total and related to grade 3/4 AEs. In the STREAM trial, AEs were graded according to the Division of AIDS, National Institute of Allergy, and Infectious Diseases criteria.

## Statistical analyses

Participants' demographic and clinical data at baseline were analyzed to assess association with DM status, using Chi-squared/Fisher's exact test for proportions or two-sample z-test/t-tests for continuous variables based on their distribution. The proportion of participants with favourable/unfavourable status, FoR event, SAE, death, AE, or unscheduled visits were calculated by DM status and compared between DM and non-DM groups using Chi-squared or Fisher's exact tests where appropriate. Regimen-wise comparison of outcomes by DM status was done if there were at least 20 events. Time-to-event endpoints (FoR and culture conversion) were analysed using Cox proportional hazard regression model (stratified by trial stage and regimen, if appropriate) to estimate hazard ratio between the DM and non-DM groups. All baseline factors considered potentially related to outcome as well as

a-priori variables were included in a multivariable model and the best model selected using Akaike Information Criterion (AIC). The proportional-hazards assumption was tested using Schoenfeld residuals and found to hold for all analyses. All analyses, except for proportion with favourable/unfavourable status are reported on a pooled basis, for Stage 1 and 2 participants combined. Proportions of participants with favourable/unfavourable status are reported separately for each trial stage because of differing primary endpoints (week 132 for stage 1 and week 76 for stage 2). All reported tests are two-sided, and analyses were conducted using STATA version 17.0.

## Determinants of glucose control

Using longitudinal glucose measures for Stage 2 participants, we performed additional analyses, logistical as well as longitudinal modelling, to evaluate determinants of glucose control. Here we did not evaluate whether poor glycemic control was associated with poor treatment outcomes. Stage 1 participants were not included in this analysis because they had a single blood glucose measurement, taken at baseline. Glucose level was considered controlled if random glucose level was <140 mg/dl. Two analyses were considered – one at 8 weeks post-randomization and the other at 16 weeks post-randomization – to cover the end of intensive phase treatment for the different regimens. At each time-point, participants were categorized as controlled/not controlled.

A logistic model to identify factors associated with glucose control was fitted using demographic, baseline clinical characteristics, and other treatment factors. In addition, longitudinal glucose measures were summarized by visit and DM status and modelled using a linear mixed model by DM status, as glycaemic control appeared to be different in the two groups. Multiple functional forms for time were considered. The best fitting model was a piecewise exponential model with knot at 20 weeks; 20 weeks was chosen as this was the best fit based on a simple plot of mean glucose level over time and made clinical sense since this coincided with the end of intensive phase of treatment. Tests of interaction between demographic and baseline clinical characteristics with time were conducted to determine variables associated with glucose control.

## Ethics Statement

The Union Ethics Advisory Group (EAG) approved (i) the use of deidentified data from the STREAM trials for this post-hoc secondary analysis, and (ii) the waiver of requirement for additional consent from participants **(EAG 53/19).**

## Results

### Efficacy Analysis population characteristics

Of the 1012 participants enrolled to the STREAM trial, a total of 896 participants were included in the efficacy analyses described here [116 were excluded either because they were not in the mITT populations (n=86), or their DM status could not be determined because of missing baseline glucose values]. Of these, majority were from Stage 2 (60%), male (62%), < 35 years (57%), self-reported as never having smoked (64%), and presented chest cavitation (76%) (Table 1).

A total of 84 (9.4%) participants had DM (per definition earlier). The mean random blood glucose (SD) at baseline was 263.2 (114.0) mg/dL in the DM group and 95.5 (19.1) mg/dl in the non-DM group (p<0.001). The proportion of the population from the two trial stages and various treatment regimens was similar in both the DM group and the non-DM group. There

**Table 1. Demographics, Baseline characteristics between DM and non-DM in efficacy analysis.**

| | Total<br>N=896 | Non-DM<br>N=812 | DM<br>N=84 | P-value |
|---|---|---|---|---|
| *Glucose (mg/DL), mean (SD)* | | 95.5 (19.1) | 263.2 (114.0) | <0.001 |
| *Stage of Trial, no. (%)* | | | | |
| Stage 1 | 357 (40) | 322 (40) | 35 (42) | 0.93 |
| Stage 2 | 539 (60) | 490 (60) | 49 (58) | |
| *Regimens, no. (%)* | | | | |
| Long | 147 (16) | 132 (16) | 15 (18) | 0.99 |
| Short | 422 (47) | 382 (47) | 40 (48) | |
| Oral | 195 (22) | 179 (22) | 16 (19) | |
| Six-month | 132 (15) | 119 (15) | 13 (15) | |
| *Country, no. (%)* | | | | |
| Ethiopia | 176 (20) | 168 (21) | 8 (10) | <0.001 |
| Georgia | 32 (4) | 30 (4) | 2 (2) | |
| India | 138 (15) | 99 (12) | 39 (46) | |
| Moldova | 59 (7) | 59 (7) | 0 | |
| Mongolia | 151 (17) | 141 (17) | 10 (12) | |
| South Africa | 192 (21) | 184 (23) | 8 (10) | |
| Uganda | 54 (6) | 54 (7) | 0 | |
| Vietnam | 94 (10) | 77 (9) | 17 (20) | |
| *Male, no. (%)* | 551 (62) | 491 (60) | 60 (71) | 0.049 |
| *Age (years), no. (%)* | | | | |
| 15 – 24 | 193 (21) | 191 (24) | 2 (2) | <0.001 |
| 25 – 34 | 321 (36) | 314 (39) | 7 (8) | |
| 35 – 44 | 203 (23) | 170 (21) | 33 (39) | |
| 45 + | 179 (20) | 137 (17) | 42 (50) | |
| BMI Category (kg/m2), no. (%) | | | | |
| Severely underweight (< 16) | 117 (13) | 116 (14) | 1 (1) | <0.001 |
| Underweight (16 – 18.49) | 267 (30) | 252 (31) | 15 (18) | |
| Normal (18.5 – 24.99) | 448 (50) | 391 (48) | 57 (68) | |
| Overweight (≥ 25) | 64 (7) | 53 (7) | 11 (13) | |
| *HIV+, no. (%)* | 191 (21) | 185 (23) | 6 (7) | 0.001 |
| Smoking status, no. (%) | | | | |
| Never smoked | 570 (64) | 515 (63) | 55 (65) | 0.01 |
| Current smoker | 153 (17) | 147 (18) | 6 (7) | |
| Ex-smoker | 173 (19) | 150 (18) | 23 (27) | |
| *\*Extent of opacities, no./total (%)* | | | | |
| None | 1/844 (0) | 1/764 (0) | 0/80 (0) | 0.48 |
| Minimal | 90/844 (11) | 85/764 (11) | 5/80 (6) | |
| Moderate | 469/844 (55) | 418/764 (55) | 51/80 (64) | |
| Advanced | 284/844 (34) | 260/764 (34) | 24/80 (30) | |
| *\*Number of cavities, no./total (%)* | | | | |
| None | 206/844 (24) | 188/764 (25) | 18/80 (23) | 0.31 |
| Single | 140/844 (17) | 121/764 (16) | 19/80 (24) | |
| Multiple | 498/844 (59) | 455/764 (60) | 43/80 (54) | |

*\*Variables with participants missing data at baseline. 52 participants were missing baseline chest X rays and therefore data for extent of opacity and number of cavities.*

was an association between DM status and country (p<.001); India and Vietnam, combined, accounted for a majority of the participants in the DM group (66%).

Participants in the DM group were more likely to be older (p<0.001); the DM group had 2-3 times the proportion of participants in the 25-34 and 35-44 age groups, respectively, compared to the non-DM group. The DM group also had a higher BMI on average compared to non-DM participants (p<0.001) and had nearly double the proportion of participants in the 'Overweight' (>25) category compared to the non-DM group. There was also borderline significant evidence (p=0.049) of an association between sex and DM status; men were more likely to have diabetes compared to women. There was an association between DM status and HIV status (p=0.001), with DM participants less likely to be HIV+ compared to non-DM; possibly explained by varying prevalence of HIV and DM between countries. There was no association between DM status and smoking history, or extent of baseline disease as determined by the radiographic extent of cavitation and lung opacities.

## Efficacy outcomes

Efficacy outcomes in the DM and non-DM groups are presented in Table 2.

The proportion of participants with an unfavourable trial outcome in the DM group was comparable to the non-DM group in both Stage 1 (unadjusted difference -6.0%, 95% CI: -21.1 to 9.2, p=0.41) and Stage 2 participants (unadjusted difference 3.9%, 95% CI: -7.1 to 14.8, p=0.52). Since this trial primary outcome was a composite endpoint that also included treatment changes, a more clinical/TB-related outcome (FoR) was also assessed. The proportion of participants who experienced definite/probable Failure or Recurrence (FoR) events were also comparable between the DM (8.3%) and the non-DM groups (5.9%); adjusted HR 1.38, 95% CI: 0.51, 3.73). Likewise, the time to culture conversion was also similar in both groups

**Table 2. Efficacy outcomes in DM and non-DM participants.**

|  | Non-DM<br>N=812 | DM<br>N=84 | P-value |
|---|---|---|---|
| *Favourable/Unfavourable status, no. (%)* |  |  |  |
| **Stage 1 – status at Week 132** | *N=322* | *N=35* |  |
| *Favourable* | 248 (77) | 26 (74) | 0.34 |
| *Unfavourable* | 61 (19) | 9 (26) |  |
| *Non-assessable* | 13 (4) | 0 |  |
| **Stage 2 – status at Week 76** | *N=490* | *N=49* |  |
| *Favourable* | 391 (80) | 41 (84) | 0.52 |
| *Unfavourable* | 99 (20) | 8 (16) |  |
| *Failure or Recurrence (FoR)* |  |  |  |
| FoR events (definite/probable), no. (%) | 48 (5.9) | 7 (8.3) | 0.38 |
| Time to FoR, unadjusted HR (95% CI) | 1.00 | 1.40 (0.63, 3.09) | 0.41 |
| Time to FoR, stratified* HR (95% CI) | 1.00 | 1.11 (0.48, 2.58) | 0.80 |
| Time to FoR, adjusted** HR (95% CI) | 1.00 | 1.38 (0.51, 3.73) | 0.53 |
| *Culture conversion* |  |  |  |
| Time to culture conversion, unadjusted HR (95% CI) | 1.00 | 0.94 (0.74, 1.18) | 0.57 |
| Time to culture conversion, stratified* HR (95% CI) | 1.00 | 0.87 (0.68, 1.11) | 0.28 |
| Time to FoR, adjusted** HR (95% CI) | 1.00 | 0.90 (0.69, 1.18) | 0.45 |

*Stratified by country and trial stage.

**Adjusted for treatment regimen, sex, age, HIV status, BMI category and stratified by country and trial stage.

(adjusted HR 0.90, 95% CI: 0.69, 1.18). None of the differences were statistically significant indicating similar efficacy outcomes in both the groups.

## Safety outcomes

A total of 976 participants were included in the safety analysis reported here (Table 3).

A significantly higher proportion of participants in the DM group experienced SAEs, excluding deaths, than in the non-DM group (41% vs. 22%, p<0.001). One-fifth of the SAEs (20.6%) in the DM group were classified as surgical/medical procedures (12.7%) or as endocrine issues (7.9%) in comparison to less than 3% in the non-DM group (S4 Table) There was also a suggestion of regimen-based differences in SAEs; proportion of SAEs in the DM and non-DM groups were comparable in the 'Long' regimen (32% DM vs. 31% non-DM) but were higher in the DM group in the 'Six-month' (62% DM vs. 17% non-DM, p<0.001), 'Oral' (29% DM vs. 15% non-DM group, p=0.123), and 'Short' (44% DM vs. 23% non-DM, p=0.002) regimens (S4 Table).

Nevertheless, the proportion of participants in the DM vs. non-DM group who died (7% vs. 5%), experienced grade 3–5 AEs (63% vs. 54%) or and had unscheduled clinic visits (68% vs. 65%), including visits reported as related to grade 3/4 AEs (14% vs. 10%) were all comparable (Table 3).

## Determinants of glucose control

Longitudinal glucose measures from 539 Stage 2 participants (490 non-DM and 49 DM) were analysed to determine determinants of glycaemic control in this population. We do not report effect of glycemic control on treatment outcomes.

Over two-thirds of DM participants (73%) were on medication for diabetes; over half (59%) on metformin and one-third on insulin (33%). In both DM and non-DM participants, blood glucose levels decrease in the first few months before plateauing (S1 Fig). Demographic characteristics of participants included in this glycemic control analysis were similar to the overall efficacy analysis population described earlier (S1 Table). There was an association between DM status and country (p <0.001), age category (p<0.001), and BMI (p<0.001). The majority of DM participants in this analysis were from India and Mongolia combined (96%), and none of them were HIV positive.

At 8 weeks post randomization, nearly all participants in the non-DM group (463/478, 97%) experienced glycaemic control (defined as glucose level <140 mg/dl). The odds of poor glycaemic control were higher in males (OR 4.55; 95% CI 1.01, 20.00) and in older

**Table 3. Safety outcomes in DM and non-DM participants.**

| | Non-DM N=889 | DM N=87 | p-value |
|---|---|---|---|
| *SAEs, excluding deaths* No. participants experiencing event (%) | 193 (22) | 36 (41) | <0.001 |
| *Deaths* No. participants (%) | 43 (5) | 6 (7) | 0.40 |
| *Grade 3-5 AEs* No. participants experiencing event (%) | 481 (54) | 55 (63) | 0.10 |
| *Unscheduled visits – total* No. participants experiencing event (%) | 579 (65) | 59 (68) | 0.62 |
| *Unscheduled visits related to AEs* No. participants experiencing event (%) | 90 (10) | 12 (14) | 0.29 |

**Table 4. Odds ratio estimates for association with poor glycaemic control from univariable logistic models (Stage 2 population only).**

| | 8 weeks post randomisation | | | | 16 weeks post randomisation | | | |
|---|---|---|---|---|---|---|---|---|
| | Non-DM (n=478) | | DM (n=46) | | Non-DM (n=466) | | DM (n=45) | |
| | OR estimate (95% CI) | P-value | OR estimate (95% CI) | P-value | OR estimate (95% CI) | P-value | OR estimate (95% CI) | P-value |
| *Treatment regimen* | | | | | | | | |
| Long | 1.84 (0.20, 17.22) | 0.59 | * | – | * | – | * | – |
| Short | 1 | – | 1 | – | 1 | – | 1 | – |
| Oral | 2.26 (0.68, 7.50) | 0.18 | 4.00 (0.39, 40.79) | 0.24 | 1.98 (0.36, 10.94) | 0.44 | 0.24 (0.04, 1.51) | 0.13 |
| Six-month | 0.34 (0.04, 3.11) | 0.34 | 0.49 (0.10, 2.40) | 0.38 | 0.70 (0.06, 7.83) | 0.77 | 0.13 (0.02, 0.86) | 0.034 |
| *Male* | 4.55 (1.01, 20) | 0.049 | 1.11 (0.24, 5.26) | 0.89 | 1.69 (0.33, 9.09) | 0.53 | 1.37 (0.36, 5.26) | 0.64 |
| *Age (years)* | | | | | | | | |
| 15 – 24 | 0.57 (0.06, 5.57) | 0.63 | – | – | * | – | – | – |
| 25 – 34 | 1 | – | 1 | – | 1 | – | 1 | – |
| 35 – 44 | 4.07 (0.99, 16.62) | 0.05 | 1.67 (0.13, 22.00) | 0.70 | 2.98 (0.49, 18.17) | 0.24 | 0.52 (0.05, 6.09) | 0.61 |
| 45 + | 4.16 (0.97, 17.83) | 0.06 | 1.00 (0.09, 11.52) | 1.00 | 2.41 (0.33, 17.41) | 0.38 | 0.76 (0.07, 8.66) | 0.83 |
| *BMI Category (kg/m²)* | | | | | | | | |
| Severely underweight (<16) | 0.67 (0.14, 3.15) | 0.62 | * | – | * | – | * | – |
| Underweight (16–18.49) | 0.33 (0.07, 1.52) | 0.15 | 0.58 (0.09, 3.88) | 0.58 | 0.28 (0.03, 2.33) | 0.24 | 2.32 (0.23, 23.42) | 0.48 |
| Normal (18.5–24.99) | 1 | – | 1 | – | 1 | – | 1 | – |
| Overweight (>25) | 0.71 (0.09, 5.70) | 0.74 | 2.04 (0.21, 19.53) | 0.54 | * | – | 1.16 (0.24, 5.58) | 0.86 |
| *HIV+* | 0.80 (0.18, 3.60) | 0.77 | – | – | * | – | – | – |
| *Smoking status* | | | | | | | | |
| Never smoked | 1 | – | 1 | – | 1 | – | 1 | – |
| Current smoker | 1.36 (0.40, 4.62) | 0.62 | * | – | 1.33 (0.24, 7.38) | 0.74 | * | – |
| Ex-smoker | 1.71 (0.44, 6.63) | 0.44 | 0.21 (0.04, 1.07) | 0.06 | 1.14 (0.44, 1.29) | 0.91 | 0.73 (0.14, 3.80) | 0.71 |
| *Smear* | | | | | | | | |
| No AFB Seen/ Rare AFB | * | – | * | – | * | – | 0.44 (0.02, 9.03) | 0.60 |
| 1+ | 1 | – | 1 | – | 1 | – | 1 | – |
| 2+ | 1.31 (0.31, 8.02) | 0.77 | 2.60 (0.39, 17.16) | 0.32 | * | – | 1.33 (0.26, 6.83) | 0.73 |
| 3+ | 2.43 (0.52, 11.34) | 0.26 | 1.10 (0.22, 5.61) | 0.91 | 0.60 (0.13, 2.74) | 0.51 | 0.59 (0.12, 2.89) | 0.52 |
| *Number of cavities* | | | | | | | | |
| None | * | – | 0.67 (0.04, 12.27) | 0.79 | * | – | 1.0 (0.16, 6.25) | 1.00 |
| Single | 1 | – | 1 | – | 1 | – | 1 | – |
| Multiple | 3.85 (0.49, 30.10) | 0.20 | 0.14 (0.01, 1.25) | 0.08 | 0.78 (0.15, 4.11) | 0.77 | 1.00 (0.22, 4.50) | 1.00 |

- No participants in this group, OR not estimated.

*Perfectly predicts outcome, small numbers of participants.

participants (OR 4.07; 95% CI 0.99, 16.62 in 35-44 years & OR 4.16; 95% CI 0.97, 17.83 in 45+ years) (Table 4). At 8 weeks, only one-fifth of participants in the DM group (10/46, 22%) experienced glycemic control and there was no evidence of association with any baseline factor.

At 16 weeks post randomization, the proportion of participants in the DM group with glycaemic control increased (15/45, 33%) with the odds of poor glycaemic control being lower in the "Six-month" regimen (OR 0.13; 95% CI 0.02, 0.86) and the "Oral" regimen (OR 0.24; 95% CI 0.04, 1.51) compared to the "Short" regimen in this group (Table 4).

In a (piecewise) longitudinal model of glucose control over time, associations found were similar to the Week 8 analysis reported above for the non-DM population – glucose control over time was worse in older (p=0.04) and male (p=0.09) participants (S2 Table & S2 Fig). No such associations were found in the DM population (S3 Table & S3 Fig).

## Discussion

There is incomplete and mixed evidence about the impact of DM on treatment and safety outcomes in TB. Several studies in people with drug-sensitive TB report poorer outcomes such as lower rates of treatment success,[21,23] culture conversion,[23] long-term survival[7,17,23] and higher rates of relapse[6,7,21], death and adverse events[7,23] in those with DM compared to those without. Studies in MDR-TB are limited but in general point to poorer outcomes in people with DM co-morbidity such as more adverse events[12], and lower rates of treatment success[10], culture conversion[11] or long-term survival[10].

The results of our study, however, provide evidence that it may be possible to achieve similar treatment outcomes in people with MDR-TB and DM co-morbidity, as those without DM. It is likely that better clinical oversight may have contributed to comparable TB outcomes between DM and non-DM in this clinical trial cohort. STREAM participants had clinic visits weekly for the first month, and then monthly for up to at least a year. Frequency of these visits likely resulted in better monitoring of participants' blood glucose which may in turn have contributed to better DM management and TB outcomes. This is consistent with some studies in drug-sensitive TB where close monitoring and good glucose control resulted in similar TB outcomes in DM and non-DM groups. In a well-resourced setting where glucose levels were closely monitored, DM was not an independent predictor of unfavorable outcomes[31]; diabetics with optimal levels of HbA1c (<7%) were more likely to have better microbiological outcomes[17]. There are several strengths to the analyses presented here from a randomised controlled trial. The study population was clearly defined with minimal loss to follow-up, data analysed was prospectively collected through standardized assessments, and confounding factors could be identified and adjusted for in the time-to-event analyses.

Nevertheless, the results presented here should be interpreted cautiously as the small sample size (84 and 87 DM participants in efficacy and safety respectively) may not have allowed detection of differences. Further, the exposure (DM status) was determined based on single random glucose measurements. A proportion of these glucose elevations may be a result of temporary dysglycaemia, commonly associated with TB disease, that auto-resolves with immune recovery following TB treatment. [2]Profile plots from the Stage 2 repeated measures analyses (S1 Fig) point to transient hyperglycaemia in all participants (DM and non-DM) early on with a decrease and plateauing of glucose levels following TB treatment and resolution of disease. In these plots, however, there is a clear distinction in glucose levels between the two groups suggesting limited misclassification using this surrogate marker for determining diabetes in this population.

Our analyses with longitudinal glucose measures also offer some valuable insights into glucose control in the Stage 2 population. While only 22% in the DM group experienced glycemic control (defined as RBG <140 mg/dl) at 8 weeks, as many as 30% and 54% had RBG levels below 154mg/dl and 200mg/dL respectively (corresponding to HBA1c levels of <7% and 9%). By 16 weeks, this increased to nearly half (42%) with optimal RBG levels (<154mg/dl), and to nearly three-fourths (71%) with RBG levels below threshold for being categorized as DM (<200mg/dl). The extent of glycemic control achieved in the DM population under this trial's setting probably explains the outcomes reported.

Responding to previous recommendations to study glycemic trajectories in MDR-TB cohorts, our analyses elucidate factors associated with glycemic control in the STREAM population [20]. Older males had worse glycemic control than their younger female counterparts in the non-DM group; this can be useful in designing clinical monitoring and management strategies.

However, given the limited number of participants in the DM group, we were not able to draw meaningful conclusions regarding factors associated with glycemic control in the DM group which needs to be further explored. Additionally, we did not evaluate whether poor glycemic control was associated with poor treatment outcomes. However, we do report better glycemic control in the DM population randomised to the "Six-month" and "Oral" regimens; intervention regimens for which superior efficacy outcomes have been reported. [28] Future studies with larger MDR-TB and DM cohorts may be able to generate further evidence around this by modelling the effect of glycemic trajectories on treatment outcomes.

Although improved treatment options for MDR-TB are now available, they are not tested or optimised for people with MDR-TB and co-morbidities such as DM. Treatment guidance and management of MDR-TB in people with and without DM co-morbidity remain the same despite several reports of poorer outcomes in those with uncontrolled diabetes [18,19]. Few trials of new drugs/regimens enrol people with diabetes and report treatment outcomes in this sub-population. Evidence needs to be generated on the efficacy of newer regimens in people with MDR-TB and DM that further informs their clinical management [13].

Our results add to the evidence on treatment outcomes using new regimens in people with MDR-TB/DM co-morbidity and support previous recommendation of use of closer monitoring and oversight to improve outcomes [32]. They also offer a promising insight that proper management of both diseases could possibly obviate the need for additional drugs or a different treatment regimen for this sub-population. While the DM group experienced more SAEs, this may be attributed to higher number of hospitalizations (S4 Table), regardless of severeness or relatedness either to TB or Diabetes, as per its definition in the conduct of clinical trials.

It would be important to evaluate the recommendations on monitoring and oversight presented here in more pragmatic settings and assess effectiveness of newer MDR-TB regimens in people with diabetes. Future research can be designed using more reliable glucose measures and advanced statistical methods/modelling techniques to describe glycemic trajectories, and to evaluate their effect on outcomes, including assessing the role of treatments for both MDR-TB and DM. Together, these can further the development of guidelines for the specific management of DM and MDR-TB co-morbidities.

## Supporting information

**S1 Fig. Glucose-level profile in glycemic control analysis population.**
(TIF)

**S1 Table. Baseline characteristics of glycemic control analysis population.**
(DOCX)

**S2 Table. Final piecewise longitudinal model for glucose levels (non-DM group).**
(DOCX)

**S2 Fig. Final piecewise longitudinal model for glucose levels (non-DM group).**
(TIF)

**S3 Table. Final piecewise longitudinal model for glucose levels (DM group).**
(DOCX)

**S3 Fig. Final piecewise longitudinal model for glucose levels (DM group).**
(TIF)

**S4 Table. Serious adverse events profiles.**
(DOCX)

## Acknowledgements

Dr A Bayissa MD, Armauer Hansen Research Institute (AHRI), Addis Ababa, Ethiopia; Dr AK Bhatnagar MD, Rajan Babu Institute for Pulmonary Medicine & Tuberculosis, Delhi, India; Dr F Conradie MBBCh, Empilweni TB Hospital, Eastern Cape, South Africa; Dr P-T Dat MD, Pham Ngoc Tach Hospital, Ho Chi Minh City, Vietnam; Dr N Gopalan DNB(chest), ICMR-National Institute for Research in Tuberculosis, Chennai, India; Dr B Kirenga PhD, Makerere University Lung Institute, Mulago Hospital, Kampala, Uganda; Prof N Kiria PhD, National Center for Tuberculosis and Lung Diseases, Tbilisi, Georgia; Dr D Meressa MD, St. Peter's Tuberculosis Specialized Hospital and Global Health Committee, Addis Ababa, Ethiopia; Dr R Moodliar MMed, THINK: Tuberculosis & HIV Investigative, Doris Goodwin Hospital, Pietermaritzburg, South Africa; Dr N Ngubane MBChB, King Dinuzulu Hospital Complex, Durban, South Africa; Dr M Rassool MBChB, Clinical HIV Research Unit, Helen Joseph Hospital, Department of Internal Medicine, University of the Witwatersrand, Johannesburg, South Africa; Dr R Solanki MD, B.J. Medical College, Ahmedabad, India; Dr B Tsogt PhD, Mongolian Anti-Tuberculosis Coalition, Ulaanbaatar, Mongolia; Dr E Tudor PhD, Institute of Phthisiopneumology "Chiril Draganiuc", Chisinau, Republic of Moldova; Dr G Torrea PhD, Institute of Tropical Medicine, Antwerp, Belgium; Prof CY Chiang PhD, Department of Internal Medicine, Wan Fang Hospital, Taipei, Taiwan, Taiwan School of Medicine, Taipei Medical University, Taipei, Taiwan and International Union against Tuberculosis and Lung Disease, Paris, France.

## Author contributions

**Conceptualization:** Meera Gurumurthy, Narendran Gopalan, Leena Patel, Gay Bronson.

**Data curation:** Gay Bronson.

**Formal analysis:** Meera Gurumurthy, Andrew Davis, Ruth Goodall.

**Methodology:** Meera Gurumurthy, Leena Patel, Andrew Davis, Ruth Goodall, Gay Bronson.

**Project administration:** Gay Bronson.

**Software:** Andrew Davis, Ruth Goodall.

**Supervision:** Gay Bronson.

**Writing – original draft:** Meera Gurumurthy, Narendran Gopalan, Gay Bronson.

**Writing – review & editing:** Narendran Gopalan, Leena Patel, Andrew Davis, Vignes Anand Srinivasalu, Shakira Rajaram, Ruth Goodall, Gay Bronson.

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
