## [Decision Letter · Decision Letter 0]

15 Sep 2024

PGPH-D-23-02559

Treatment outcomes in people with diabetes and multidrug-resistant tuberculosis (MDR TB) enrolled in the STREAM clinical trial.

Dear Dr. Gurumurthy,

Thank you for submitting your manuscript to PLOS Global Public Health. After careful consideration, we feel that it has merit but does not fully meet PLOS Global Public Health’s publication criteria as it currently stands. Therefore, we invite you to submit a revised version of the manuscript that addresses the points raised during the review process.

We look forward to receiving your revised manuscript.

Kind regards,

Joel Msafiri Francis, MD, MS, PhD

Academic Editor

Additional Editor Comments (if provided):

Reviewers' comments:

Reviewer's Responses to Questions

**Comments to the Author**

1. Does this manuscript meet PLOS Global Public Health’s publication criteria ? Is the manuscript technically sound, and do the data support the conclusions? The manuscript must describe methodologically and ethically rigorous research with conclusions that are appropriately drawn based on the data presented.

Reviewer #1: Partly

Reviewer #2: Yes

Reviewer #3: Yes

2. Has the statistical analysis been performed appropriately and rigorously?

Reviewer #1: Yes

Reviewer #2: Yes

Reviewer #3: Yes

3. Have the authors made all data underlying the findings in their manuscript fully available (please refer to the Data Availability Statement at the start of the manuscript PDF file)?

Reviewer #1: Yes

Reviewer #2: Yes

Reviewer #3: Yes

4. Is the manuscript presented in an intelligible fashion and written in standard English?

Reviewer #1: Yes

Reviewer #2: Yes

Reviewer #3: Yes

5. Review Comments to the Author

Reviewer #1: In this manuscript, the authors present a secondary analysis to determine the impact of comorbid diabetes on MDR-TB treatment outcomes. Like many to large cohorts, they attempt to draw power from a minority exposure within a large data set. In their evaluation, they did not identify significant differences in treatment outcomes, however they report non-significant outcomes as suggestive of differences being present, which is not consistent with the underlying data. While the finding that the absence of a significant difference between study groups may warrant presentation, the manner in which is is represented is a bit misleading. Overall, there are challenges in both the description of the exposure and the outcome as outlined below, which limits the extent to which these data as presented contribute to the literature.

Major Comments:

Non-significant findings should be presented as not being different, rather than representing differences that were not significant.

The study would be improved by evaluation of their exposure of interest using glycosylated hemoglobin, rather than single random glucose measurements, as that may more accurately reflect the distribution of diabetes within this population.

The authors note that they included change in treatment regimen as an unfavorable outcome, which is extremely common during MDR-TB treatment in order to optimally manage toxicities. While this is commonly represented in studies of DS-TB, often indicating emerging resistance, if the authors were to present this as a primary outcome, it should be presented in parallel with an analysis of unfavorable MDR-TB outcomes that exclude change in regimen.

Finally, the authors fail to account for the dysglycemia of chronic illness commonly associated with TB disease, much of which self-resolves with immune recovery following TB treatment. As such, the presence of for example, an A1c of 6.6% resolving to 6.3%, might represent someone who is not truly diabetic beyond this temporary dysglycemia, and the analysis and discussion presented in the manuscript would need to take this into account.

Reviewer #2: The review is well written.

I found no issue related to the statistical analyses performed in the study.

The ethical considerations have been presented.

The tables are well presented and the numbers are correct.

The conclusion is in line with the findings of the study.

Reviewer #3: 1. The Abstract looks too long; can it be somewhat tightened?

2. The statistical analyses involved well known techniques, such as Fisher's exact test, Chi-squared test (for proportions), and 2-sample z-tests/t-tests. It also involves fitting proportional hazards regression with time-to-event endpoints. If I am not mistaken, I do not see the assessment of the proportional hazards assumptions, which can be easily done using standard software, such as SAS, R, STATA, etc.

3. A piecewise longitudinal model of glucose control over time was fitted. This fitting was not mentioned in the statistical analysis section. Furthermore, the justifications behind the piecewise construction is not provided clearly.

6. PLOS authors have the option to publish the peer review history of their article (what does this mean? ). If published, this will include your full peer review and any attached files.

**Do you want your identity to be public for this peer review?** For information about this choice, including consent withdrawal, please see our Privacy Policy .

Reviewer #1: No

Reviewer #2: **Yes: ** Jean Pierre Fina Lubaki

Reviewer #3: No

---

## [Decision Letter · Decision Letter 1]

17 Jan 2025

Treatment outcomes in people with diabetes and multidrug-resistant tuberculosis (MDR TB) enrolled in the STREAM clinical trial.

PGPH-D-23-02559R1

Dear Dr Gurumurthy,

We are pleased to inform you that your manuscript 'Treatment outcomes in people with diabetes and multidrug-resistant tuberculosis (MDR TB) enrolled in the STREAM clinical trial.' has been provisionally accepted for publication in PLOS Global Public Health.

Best regards,

Joel Msafiri Francis, MD, MS, PhD

Academic Editor

Reviewer Comments (if any, and for reference):

Reviewer's Responses to Questions

**Comments to the Author**

1. If the authors have adequately addressed your comments raised in a previous round of review and you feel that this manuscript is now acceptable for publication, you may indicate that here to bypass the “Comments to the Author” section, enter your conflict of interest statement in the “Confidential to Editor” section, and submit your "Accept" recommendation.

Reviewer #1: All comments have been addressed

Reviewer #2: All comments have been addressed

Reviewer #3: All comments have been addressed

2. Does this manuscript meet PLOS Global Public Health’s publication criteria ? Is the manuscript technically sound, and do the data support the conclusions? The manuscript must describe methodologically and ethically rigorous research with conclusions that are appropriately drawn based on the data presented.

Reviewer #1: Yes

Reviewer #2: Yes

Reviewer #3: (No Response)

3. Has the statistical analysis been performed appropriately and rigorously?

Reviewer #1: Yes

Reviewer #2: Yes

Reviewer #3: (No Response)

4. Have the authors made all data underlying the findings in their manuscript fully available (please refer to the Data Availability Statement at the start of the manuscript PDF file)?

Reviewer #1: Yes

Reviewer #2: Yes

Reviewer #3: (No Response)

5. Is the manuscript presented in an intelligible fashion and written in standard English?

Reviewer #1: Yes

Reviewer #2: Yes

Reviewer #3: (No Response)

6. Review Comments to the Author

Reviewer #1: The authors have adequately addressed my previous comments, though the acknowledgement of negative findings may adjust the expected impact of the submission.

Reviewer #2: (No Response)

Reviewer #3: (No Response)

7. PLOS authors have the option to publish the peer review history of their article (what does this mean? ). If published, this will include your full peer review and any attached files.

**Do you want your identity to be public for this peer review?** For information about this choice, including consent withdrawal, please see our Privacy Policy .

Reviewer #1: No

Reviewer #2: **Yes: ** Jean-Pierre FINA LUBAKI

Reviewer #3: No
